# Diffusion Coefficients and Activation Energies of Diffusion of Organic Molecules in Polystyrene below and above Glass Transition Temperature

**DOI:** 10.3390/polym13081317

**Published:** 2021-04-16

**Authors:** Frank Welle

**Affiliations:** Fraunhofer Institute for Process Engineering and Packaging (IVV) Giggenhauser Straße 35, 85354 Freising, Germany; frank.welle@ivv.fraunhofer.de

**Keywords:** polystyrene, activation energy, diffusion coefficients, functional barrier, diffusion modelling

## Abstract

General Purpose Polystyrene (GPPS) and High Impact Polystyrene (HIPS) is used in packaging food as well as for technical products. Knowledge of the diffusion behavior of organic molecules in polystyrene (PS) is important for the evaluation of the diffusion and migration process. Within this study, diffusion coefficients were determined in GPPS and HIPS below and above the glass transition temperature. Diffusion coefficients were determined from desorption kinetics into the gas phase using spiked GPPS and HIPS sheets as well as from permeation kinetics through a thin GPPS film. Overall, 187 diffusion coefficients were determined in GPPS and HIPS at temperatures between 0 °C and 115 °C. From the temperature dependency of the diffusion coefficients 45 activation energies of diffusion E_A_ and the pre-exponential factor D_0_ were determined. As expected, the activation energies of diffusion E_A_ show a strong dependency from the molecular volume of the investigated substances. At the glass transition temperature, only a slight change of the diffusion behavior were observed. Based on E_A_ and D_0_, prediction parameters for diffusion coefficients were established.

## 1. Introduction

Polystyrene (PS) is widely used as packaging materials for food [1]. Crystal polystyrene (general purpose polystyrene (GPPS)) is used for containers for a variety of foods and as disposable cups for beverages. GPPS is normally not suitable for packaging of food with a high fat content, e.g., salad dressings and margarine, because the high fat levels may cause stress cracking resulting in a decreased barrier function [2]. To overcome the brittleness of polystyrene, butadiene synthetic rubbers (between 5% and 12%) are added during polymerization to manufacture high impact polystyrene (HIPS). Opaque HIPS is used for food containers and yogurt cups. GPPS and HIPS plastics have poor barrier properties to permanent gases like oxygen and carbon dioxide. This reduces the shelf life of food packed in PS, but on the other hand, makes HIPS a good packaging material for yogurt and milk products as some penetration of oxygen is necessary to assist the yogurt fermentation process. Foamed PS is thermoformed into a variety of trays for meat, poultry, fish, fruit and vegetables as well as containers for eggs, and fast foods, and disposable cups for hot beverages. PS is also used for technical products or applications such as electronics, furniture, toys, building insulation and automotive parts.

Residual volatile substances like residual monomers or side-products formed during the polymerization process, e.g., styrene, toluene, ethylbenzene, *iso*-propylbenzene and *n*-propylbenzene can be determined in the polystyrene packaging materials and might migrate into food after contact with polystyrene packaging [3,4,5,6,7,8]. Polymer additives [9] and oligomers [10,11,12,13] can also migrate into food. In order to predict this migration, diffusion coefficients D_P_ for organic substances in the polymer should be available. Since PS is used also for hot beverages, diffusion coefficients at high temperatures should also be available [6,14]. In this regard, activation energies of diffusion are useful for the prediction of the migration at elevated temperatures.

In some cases, PS is used as a functional barrier in order to reduce the migration of polymer additives or dyes [15,16]. Prediction of barrier properties towards organic compounds also needs knowledge about the diffusion behavior of PS. In addition, prediction of diffusion coefficients might be useful for the evaluation of post-consumer recyclates for reuse in food packaging applications as well as in cleaning efficiency simulation of the corresponding recycling processes. This had been successfully shown for polyethylene terephthalate (PET) [17,18].

To the knowledge of the author, systematic studies on diffusion coefficients D_P_ and activation energies of diffusion E_A_ for GPPS and HIPS are not available in the scientific literature. Most of the diffusion coefficients available in the scientific literature are derived from migration experiments in food or food simulants. As mentioned above, GPPS is not suitable for fatty food which interacts with the polymer, resulting in increasing migration and consequently, higher diffusion coefficients. HIPS might also show strong interactions with food simulants. Food simulants like ethanol/water mixtures, *n*-heptane or *iso*-octane show strong interactions with polystyrene leading to an over-estimation of the migration which results into too high diffusion coefficients due to swelling and stress cracks compared to the pure diffusion of organic molecules. On the other hand, most of the typical applied foods do not show such strong interactions with PS. Therefore, diffusion coefficients based on migration measurements into food simulants will lead to prediction parameters which are too conservative. In addition, most of the diffusion coefficients which are used for the prediction models are derived from single point measurements, e.g., after storage for 10 d at 40 °C or 60 °C [9]. From single point measurements, it is not possible to show if the diffusion process is following Fickian laws of diffusion. Parametrization of prediction parameters based on migration kinetics appears to be a more suitable approach for the determination of diffusion coefficients in PS or in polymers in general.

The aim of this study was to determine the diffusion coefficients of organic molecules in GPPS and HIPS over a broad temperature range below and above the glass transition temperature of 100 °C. Diffusion coefficients were determined from desorption kinetics into the gas phase using spiked GPPS and HIPS sheets as well as from permeation kinetics through a thin GPPS film. From the temperature dependency of the diffusion coefficients the activation energies of diffusion E_A_ and the pre-exponential factor D_0_ were determined. From the results, parameters for the prediction of diffusion coefficients for organic molecules in GPPS and HIPS below and above the glass transition temperature were derived.

## 2. Materials and Methods

### 2.1. Sample Materials and Model Compounds for Desorption Testing

#### 2.1.1. Manufacturing of Polystyrene Sheets Spiked with Model Compounds

Pairs of spiked sheets with thicknesses of 350 µm were prepared from GPPS and HIPS polymer for the desorption kinetics into the gas phase. Model compounds were used as surrogates for real contaminants because polystyrene samples with real contaminants in the molecular weight range of interest are rarely available. Two sets of model compounds were spiked homogeneously during polystyrene sheet production. The first sheet was spiked with a mixture of n-alkanes n-octane, n-decane, n-dodecane, up to n-tetracosane. The second sheet was spiked with a mixture of the following substances: acetone, ethyl acetate, toluene, chlorobenzene, phenyl cyclohexane, benzophenone and methyl stearate.

#### 2.1.2. Quantification of Spiking Levels in the Polystyrene Sheets

The concentrations of the model compounds were determined quantitatively in the PS sheets by extraction with acetone as solvent. A 1.0 g measure of the PS material was immersed with 10 mL acetone and stored at 40 °C for 3 d. Subsequently, the solvent was removed and the sheets were extracted again in order to prove if the first extraction was exhaustive. The extracts were then analyzed by gas chromatography with flame ionization detection (GC–FID): Column: DB 1; 20 m length; 0.18 mm inner diameter; 0.18 µm film thickness, temperature program: 50 °C (2 min), followed by heating at 10 °C/min to 340 °C (15 min), pre-pressure: 50 kPa hydrogen, split: 10 mL/min. Calibration was achieved by standard addition of the model compounds. *tert*-Butylhydroxyanisole (BHA, CAS No. 8003-24-5) and Tinuvin 234 (CAS No. 70321-86-7) were used as internal standards. The concentrations of the very volatile substances acetone and ethyl acetate were estimated from the headspace gas chromatograms (GC–FID): Column: ZB 1;30 m length; 0.25 mm inner diameter.; 0.32 µm film thickness, temperature program: 50 °C (4 min), followed by heating at 20 °C/min to 320 °C (15 min), pre-pressure: 50 kPa helium, split: 10 mL/min. Headspace Autosampler: Oven temperature: 150 °C, needle temperature: 160 °C, transfer line temperature: 170 °C, equilibration time: 1 h, pressurization time: 3 min, injection time: 0.02 min, withdrawal time: 1 min. Concentrations were estimated compared to a neat standard of the substances. The concentrations of the model compounds in the investigated PS sheets are summarized in Table 1.

For each test, 1.0 g of sample material was weighed into a headspace vial and analyzed by headspace GC–FID. Gas chromatograph: Perkin Elmer AutoSystem XL, column: ZB 1; 30 m length; 0.25 mm inner diameter; 0.32 µm film thickness, temperature program: 50 °C (4 min) followed by heating at 20 °C/min to 320 °C (15 min), pre-pressure: 50 kPa helium, split: 10 mL/min. Headspace Autosampler: Perkin Elmer HS 40 XL, oven temperature: 150 °C, needle temperature: 160 °C, transfer line: 170 °C, equilibration time: 1 h, pressurization time: 3 min, injection time: 0.02 min, withdrawal time: 1 min. Quantification of limonene was achieved by external calibration with standards of different concentrations.

#### 2.1.3. Determination of Diffusion Coefficients

Migrations into the gas phase of the spiked model compounds were determined according to [19,20] using an automated method that involved placing sheet samples of 15.6 cm diameter (area 191 cm^2^) in a migration cell. One sheet per temperature was analyzed. The migration cell was heated up to the measuring temperature and the compounds migrating from the polystyrene sheets were purged out of the extraction cell by a helium stream of 20 mL/min. The migrants were trapped (Carbopack B 20 mm, Supelco) at a trap temperature of –46 °C. Subsequently, the loaded trap was completely desorbed and transferred directly to the connected gas chromatograph (GC) by heating it to 300 °C within about 10 s. Subsequently, a new trapping cycle started. By using this automated method, every 40 min a kinetic point was determined. The migrants were separated during the GC run and quantified during the chromatographic measurements. Calibration was achieved by injection of undiluted standard solutions of the migrants into the migration cell. Gas chromatograph: Column: Rxi 624; 30 m length; 0.32 mm inner diameter; 1.8 µm film thickness, temperature program: 40 °C (2 min), rate 20 °C/min, 270 °C (8 min), pressure 70 kPa helium, detector temperature: 280 °C.

The diffusion coefficients D_P_ were calculated from the area related migration into the gas phase according to Equation (1). Within this equation, m_t_/A (in µg/cm^2^) is the mass transfer into gas phase per area A (in cm^2^) and δ (in g/cm^3^) is the density of the polymer. The concentration of the migrant at the start of the tests is c_P,0_ (in mg/kg) and D_P_ (in cm^2^/s) is the diffusion coefficient in the polymer. The parameter t is the run time (in s) of the experiment.
(1)$$\frac{m_{t}}{A}=\frac{2}{\sqrt{\pi }}\delta \;c_{P,0}\sqrt{D_{P}\;t}$$

### 2.2. Sample Materials and Model Compounds for Permeation Testing

Permeation of 1-alcohols was determined through a biaxially oriented GPPS film of 34 ± 1 µm thickness purchased from Goodfellow (Huntingdon, UK). The permeation rates were determined for 1-alcohols from methanol to 1-octanol. The homologous rows of substances with different polarities were chosen in order to establish correlations which might be useful to predict the diffusion behavior of other, non-tested substances.

### 2.3. Permeation Measurements

The 34 µm GPPS film was clamped in a stainless steel permeation cell between two sealant rings. The surface area of the tested films was 191 cm^2^. The permeation cell with the film was placed in a climate chamber. One film per temperature was analyzed. The cell has a lower and an upper space separated by the film. The lower space of the permeation cell had a volume of 7667 cm^3^ and was spiked with the permeants. The starting concentrations (c_gas phase_) of the investigated permeants (1-alcohols) in the lower space of the permeation cell, their molecular weights and molecular volumes [21] are given in Table 2. The upper space of the permeation cell was rinsed with pure nitrogen (20 mL/min) which moved the permeated substances out of the cell. The nitrogen stream went through a connected enrichment unit and the permeants were trapped on this unit. The enrichment unit was connected to a gas chromatograph with flame ionization detection (GC–FID) and the permeants were directly desorbed into the gas chromatograph. By use of this technique, the permeated amount into the upper space of the permeation cell was analyzed for the applied permeants. During the GC run, the next sample was again trapped on the enrichment unit and subsequently injected into the GC. By use of this method, kinetic points were measured every 45 min. Gas Chromatographic Conditions: Column: Rxi 624; 30 m length; 0.32 mm inner diameter; 1.8 µm film thickness, carrier gas: 120 kPa helium. Temperature program: 40 °C (2 min), rate 20 °C/min to 280 °C hold for 7 min. Pre-trap: substances collected on 20 mm length by 5 mm diameter of Carbopack 107 (Supelco), desorbed at 350 °C. Main trap: substances focused at –46 °C on 20 mm length by 1.4 mm inner diameter of Carbopack B, desorbed at 260 °C. Calibration was performed with injections of known amounts of the applied permeants.

From the experimental data, the permeation rates as well as the lag times of the applied permeants are available. The diffusion coefficient D_P_ of the applied permeants in GPPS was calculated from the lag time t_lag_ (in s) and the thickness l (in cm^2^) of the film according to Equation (2) [22,23].
(2)$$D_{P}\;=\;\frac{l^{2}}{6\;\;t_{\mathrm{lag}}}$$

### 2.4. Calculation of Molecular Volumes

The molecular volume V (in Å^3^) of the molecules was calculated with the free internet program “molinspiration” [21]. This program calculates the van der Waals volume of organic molecules. The method for calculation of molecule volume developed is based on group contributions.

## 3. Results and Discussion

Within this study, diffusion coefficients D_P_ of several organic compounds were determined by use of two independent methods: (i) Desorption kinetics from spiked GPPS and HIPS sheets and (ii) permeation kinetics on a thin (34 ± 1 µm) non-spiked GPPS film.

### 3.1. Diffusion Coefficients from Desorption Kinetics

The applied desorption method determines the migration of spiked substances from PS sheets into the gas phase at elevated temperatures. Diffusion coefficients D_P_ were determined from the slopes of the linear correlation between the square root of time and the area-dependent migration according to Equation (1). By use of this desorption method, several migrants can be determined simultaneously provided the model compounds are distributed homogenously in the spiked PS sheet and separation on the GC column is attained. In order to obtain a homogenous distribution of the migrants in the sheet, the organic substances were spiked into the polymer melt during sheet production. Notably, a proportion of the migrants evaporate during the thermal step of sheet production, especially in case of volatile substances. Thus, compounds of higher volatility are removed in greater amounts during sheet production, leading to lower concentrations in the final spiked PS sheet. It was therefore necessary to analyze the final PS sheets according to their residual concentration of the artificially spiked compounds (Table 1). As a result, the concentrations of the *n*-alkanes were quantified between 400 and 700 mg/kg. A broader concentration range was determined for the applied solvents like acetone, ethyl acetate, toluene and chlorobenzene with concentrations of 180 mg/kg up to around 800 mg/kg. The substances phenyl cyclohexane and benzophenone were determined at concentrations of 500 to 1200 mg/kg. The concentration of styrene in both sheets was determined in the range of 350 mg/kg up to 600 mg/kg. Styrene was not artificially added, but detectable in polystyrene polymers as residual monomer.

The diffusion coefficients D_P_ derived from the desorption kinetics are summarized in Table 3 (*n*-alkanes) and Table 4 (other substances with various functional groups and aromatic rings). Examples of the desorption kinetics into the gas phase for *n*-octane and chlorobenzene in GPPS as well as for *n*-dodecane and toluene in HIPS are given in Figure 1. Desorption kinetic curves all other applied substances had a similar shape.

The desorption kinetics were determined at temperatures between 75 °C and 115 °C. For all investigated migrant kinetic points, they were determined every 40 min. Therefore, several kinetic points are available for each diffusion coefficient. From this migration, it could be shown that the diffusion process is following Fickian laws of diffusion resulting in a linear correlation between the migrated amount and the square root of time [21]. However, it should be noted that the kinetic curves shown in Figure 1 do not go through the zero point. This is due to the fact that during sheet manufacturing, a portion of the spiked substances are lost from the hot surface of the sheets during sheet production, which reduces slightly the concentration at the surface of the sheets. This leads to a slightly lower desorption at the beginning of the kinetics. Due to the high temperatures during the kinetic tests, the concentration is gradually replenished at the surface of the sheets. Therefore, the lower surface concentration at the beginning of the kinetics has no influence on the measured diffusion coefficients D_P_, because the slopes after this initial phase were taken into account.

The measured diffusion coefficients were determined in the range of 10^−8^ cm^2^/s to 10^−16^ cm^2^/s depending on the substance and the applied temperature. As expected, low molecular weight molecules show significantly higher diffusion coefficients D_P_ than high molecular weight substances, especially at low temperatures. The glass transition temperature T_g_ of polystyrene of around 100 °C is within the measuring interval for both GPPS and HIPS. In the case of GPPS sheet 1 spiked with *n*-alkanes, five kinetic points were determined below T_g_ and three kinetic points were determined above T_g_ (both include the kinetic point at T_g_ of 100 °C). In case of sheet 1, it was possible to calculate the activation energies of diffusion below as well as above the glass transition temperature. Regarding GPPS sheet 2, only one kinetic point was determined above T_g_. Therefore, activation energies of diffusion are not available above the glass transition temperature and the diffusion coefficients at 105 °C were applied to determine the activation energies of diffusion below the glass transition temperature. A similar situation was available for HIPS. Four (sheet 1) and five (sheet 2) kinetic points were available below, whereas two and three kinetic points, respectively, were available above the glass transition temperature. In the case of HIPS, the determination of the activation energies of diffusion was possible below and above the glass transition temperature. However, due to only a couple of kinetic points and the small temperature interval, the activation energies of diffusion above the glass transition temperature are less precise for HIPS compared to GPPS.

### 3.2. Diffusion Coefficients from Permeation Kinetics

The permeation curves of six 1-alcohols through a 34 µm biaxially oriented GPPS film were determined in this study. The diffusion coefficients were determined from the lag times of the permeation curves according to Equation (2). The lag time t_lag_ is defined as the intercept of the asymptote to the permeation curve on the time-axis [22]. In previous studies, the same method was applied on thin films of oriented polyamide (PA6) [24], polyethylene terephthalate (PET) [23,25], polyethylene naphthalate (PEN) [26] and ethylene vinyl alcohol copolymer (EVOH) [27]. Examples for the experimental permeation curves for 1-butanol through the investigated GPPS film at temperatures between 60 °C and 90 °C are given in Figure 2. The permeation curves of the other substances measured within this study follow a similar behavior. The diffusion coefficients D_P_ for the applied 1-alcohols are summarized in Table 5. Permeation tests with *n*-alkanes failed because the film became brittle and broke under the applied temperature and concentration conditions.

The lag times and diffusion coefficients in the biaxially oriented GPPS film were determined at temperatures between 0 °C and 90 °C. However, it was only for methanol and ethanol diffusion that coefficients could be determined at temperatures of 0 °C and 25 °C, because the diffusion coefficients D_P_ of all other 1-alcohols were too low at ambient temperatures. Diffusion coefficients increase significantly with molecular volume and therefore the lag time increases accordingly. For example, given a diffusion coefficient D_P_ of 9.2 × 10^−15^ cm^2^/s, which is the expected diffusion coefficient of 1-butanol from the results of this study, a lag time of 6.6 years is predicted according to Equation (2). Raising up the temperature to 30 °C, the lag time is still around 3 years and hard to be measured in a reasonable time. Thinner GPPS films, which will lead to significantly lower lag times for larger molecules, are not available on the market and also the handling with such thin and brittle films is difficult. Therefore, higher temperatures need to be applied for 1-alcohols starting from 1-propanol. It is interesting to note that the diffusion coefficients for methanol are more or less similar in the range of 1 to 3 × 10^−9^ cm^2^/s at all applied temperatures, which indicates that the activation energy of diffusion is virtually zero. The diffusion coefficients of ethanol increase in the same temperature interval from 2.8 × 10^−11^ cm^2^/s (0 °C) to 1.6 × 10^−9^ cm^2^/s (90 °C) (Table 5).

In the permeation kinetic, the initial concentrations of the permeants in the lower cell (Table 2) were chosen such that they are a factor of approx. 200 below the saturated vapor pressure at each temperature. This avoids condensation of the permeants on the surface of the GPPS film. As a consequence, swelling of the polymer and the associated increase of the diffusion coefficients D_P_ were minimized. Without swelling of the polymer, the determined coefficients can be considered as pure diffusion coefficients in the GPPS polymer. In other trials, the permeation *n*-alkanes were also tested at similar low concentrations in the gas phase. However, after the contact of the GPPS film with the *n*-alkanes, the film becomes brittle and a breakthrough of the permeants increased significantly with a non-Fickian diffusion behavior. Therefore, the diffusion coefficients cannot reliably be derived. Diffusion coefficients for *n*-alkanes are therefore not available from permeation tests on thin GPPS films.

### 3.3. Activation Energies of Diffusion

Within this study the diffusion coefficients were determined at different temperatures between 75 and 115 °C. The activation energies of diffusion are calculated from the diffusion coefficients according to the Arrhenius approach [28]. In all cases, the Arrhenius plots show good linearity for the investigated substances. This indicates that the diffusion process follows Fickian laws and any swelling of the polymer by the permeants can be neglected under the experimental conditions applied within this study. Activation energies are only calculated when a minimum of four kinetic points are available. This serves to ensure that the values determined are sufficiently precise for using in the parameterization of the prediction parameters.

The determined diffusion coefficients show a strong dependency on the size of the migrating substance, represented by the molecular volume, as well as on temperature. As expected, for larger molecules, the diffusion coefficients are significantly lower when compared to very small molecules like acetone of methanol. In addition, lower temperatures result in lower diffusion coefficients for each individual permeant, which is in agreement with diffusion theory. From the slopes and the intercepts, the activation energies of diffusion E_A_ as well as the pre-exponential factors D_0_ were calculated. The calculated activation energies of diffusion E_A_ are given in Table 6 (GPPS) and Table 7 (HIPS). The Arrhenius plots and the correlation of the reciprocal temperature versus diffusion coefficient are given in Figure 3.

For *n*-alkanes and styrene diffusion, coefficients are available below and above the glass transition temperature T_g_ of 100 °C (Figure 3a). The results show that the diffusion behavior of GPPS and HIPS changes slightly at the glass transition temperature T_g_ (Figure 3a,b). The activation energies of diffusion E_A_ are lower above T_g_ compared to the values below T_g_, which results in a lower slope of the Arrhenius plot above T_g_. The change in the diffusion behavior is more significant for larger molecules compared to smaller molecules. Relatively small sized molecules like styrene and *n*-octane show only a slight change in the diffusion behavior whereas *n*-octadecane shows a significant change of the diffusion behavior at the glass transition temperature. In the case of GPPS sheet 2, only one temperature was determined above T_g_ (Figure 3c). The results indicate that that for larger molecules like phenyl cyclohexane and benzophenone, the activation energy of diffusion E_A_ is also lower above T_g_ compared to the slopes below T_g_, which is in agreement with the results of the *n*-alkane spiked sheet. However, due to the fact that only one temperature is available, these findings are on a weak basis. However, the results are in agreement with the finding for HIPS (Figure 3d), where three temperatures were measured above T_g_. Benzophenone shows a lower activation energy of diffusion E_A_ above T_g_.

The diffusion coefficients determined from the permeation test on the 34 µm GPPS film are determined only below the glass transition temperature at temperatures between 0 °C and 90 °C (Figure 3e). Therefore, no conclusion can be drawn regarding the diffusion behavior above T_g_ from the results of the permeation tests. In a previous study on PET, we found no significant change of the diffusion behavior at the glass transition temperature of 69 °C [19]. However, the tested substance (tetrahydrofuran, THF) is also a relatively small molecule and larger molecules might also show a change of the diffusion behavior of PET. Thus, the results of this study on PS might be consistent with the results of the previous study on PET.

### 3.4. Prediction of Diffusion Coefficients

Diffusion coefficients at various temperatures were determined in this study from which activation energies of diffusion were derived. Based on these experimentally determined activation energies, a correlation between activation energies of diffusion E_A_ and the molecular volume of the migrant V was established for GPPS and HIPS (Figure 4). Similar correlations between E_A_ and V were found in the literature for other polymers like PET [25], polycycloolefin polymer (COP) [20], polyethylene naphthalate (PEN) [26] and polyethylene vinylalcohol copolymer (EVOH) [27]. All tested compounds follow the correlation between the E_A_ and V nearly independent from chemical nature, functional groups or polarity of the molecules. Therefore, most probably also other organic molecules in PS might follow similar correlations. Due to slight effects of polarity or experimental uncertainties, the correlation of the activation energies of diffusion E_A_ and the molecular volume V shows a slight variance. As expected, the diffusion below and above the glass transition temperature is different, resulting in different correlations between the activation energies of diffusion E_A_ and the molecular volume V. As discussed above, for a given molecular volume V, the activation energies of diffusion E_A_ above T_g_ are lower than below T_g_ as indicated for the substances styrene, *n*-octane, *n*-decane, *n*-dodacane, *n*-tetradecane, and *n*-hexadecane. As shown in Figure 5, also the pre-exponential factor D_0_ correlates also with the activation energy E_A_ for GPPS and HIPS, above as well as below the glass transition temperature. The role of the correlation between the pre-exponential factor D_0_ and the activation energy E_A_ has been discussed in the literature as a so-called “compensation effect” [29] or “Meyer–Neldel rule” [30] and was also found in previous studies [20,25,26,27]. As a consequence of this effect, the activation energy E_A_ and the pre-exponential factor D_0_ are not independent from each other. This correlation was established over 16 orders of magnitude in the case of PET [25], 29 for COP [20], 23 for PEN [26], 30 for EVOH [27] and 74 orders of magnitude for PS in this study. It is noteworthy that in the case of GPPS and HIPS, the correlation does not significantly change below and above T_g_, whereas the correlation between the molecular volume V and the activation energy of diffusion E_A_ is different below and above the glass transition temperature. From the correlations given in Figure 4 and Figure 5, the activation energies of diffusion E_A_ as well as the pre-exponential factor D_0_ can be predicted for other non-tested substances. From both values, the diffusion coefficients are available at any temperature within the measured temperature intervals. Assuming that the Arrhenius plot is linear over a broader temperature range, diffusion coefficients can also be predicted at lower temperatures, e.g., room temperature. An equation was established combining both correlations and is given in Equation (3). The prediction of diffusion coefficients might be an important tool for the prediction of the mass transfer from PS into contact media, e.g., foodstuffs.

Based on the correlations shown in Figure 4 and Figure 5, the parameters *a* to *d* in Equation (3) were derived according to the procedure published in [25]. The parameters *a* to *d* for Equation (3) below and above the glass transition temperature are given in Table 8. It should be noted that the parameters for GPPS above the glass transition temperature are derived only from six individual activation energies of diffusion and pre-exponential factors, whereas the correlation below the glass transition temperature is based on 17 activation energies of diffusion. The correlation above the glass transition temperature therefore is associated with a high degree of uncertainty and the values should be used with caution. The parameters for HIPS were derived from ten activation energies of diffusion below and above the glass transition temperature.

The parameter *a* is the slope of the correlation between the activation energies of diffusion E_A_ and the pre-exponential factor D_0_ of the Arrhenius equation. When this reciprocal temperature *a* is reached, the diffusion coefficients reaches its maximum value, which is the factor *b*. The parameter *b* is therefore the highest diffusion coefficient, which can be reached in the investigated temperature interval. When the molecular volume reaches the parameter *c*, the activation energy of diffusion reaches zero and the diffusion coefficients are the same at any temperatures, which is again the parameter *b* but at the same time also D_0_ when the molecular volume V is the parameter *c*. The parameter *d* is a measure of the extent to which the activation energy of diffusion is influenced by the molecular volume V of the migrant in different polymers.
(3)$$D_{P}\;=\;b\;\;\left(\frac{V}{c}\right){\;}^{\frac{a\;-\;\frac{1}{T}}{d}}$$

For migration modelling used for food law compliance evaluation, the applied prediction models should be over-estimative, because the predicted migration is in any case higher as the experimentally determined migration [31]. As a consequence, the predicted diffusion coefficients D_P_ needs to be higher as the measured diffusion coefficients for each molecule in any case.

An artificial reduction, e.g., of 20% of the molecular volume seems to sufficient to overestimate the experimentally determined migration in case of PET [25]. These diffusion coefficients might be considered as worst-case diffusion coefficients for food law compliance evaluation. Systematic studies for polystyrene are not available to date, so that the amount of volume reduction needs to part of further investigations. On the other hand, when using migration modelling for the prediction of the decontamination efficiency in PS recycling processes, over-estimation will be the best-case. In this case, the molecular volume should be increased in order to get slightly lower diffusion coefficients D_P_ for the prediction of the worst-case cleaning efficiencies of a recycling process. In conclusion, by use of Equation (3), the diffusion coefficients of other non-tested substances in GPPS and HIPS can be predicted for any temperature between 0 °C and 115 °C if the molecular volume V is known.

Polarity of the substances seems to play a minor role on the correlations in Figure 4. The polarity and the functional groups of the molecules influences the partition coefficient K between the polymer and the gas phase. This is discussed in previous studies [24,26].

It is interesting to note that the very low molecular weight alcohols like methanol and ethanol have molecular volumes V which are smaller or near the volume of parameter *c* for GPPS. According to Equation (3), molecules with a molecular volume of parameter *c* result in an activation energy of diffusion of 0 kJ/mol and the diffusion coefficient is equivalent to factor *b*. For methanol, an activation energy of diffusion of 4.9 kJ/mol was determined which is within the analytical uncertainties in good agreement with Equation (3). For ethanol, an activation energy of diffusion of 37.3 kJ/mol was determined from the permeation tests. The activation energy of acetone was determined to 7.6 kJ/mol, which was also in good agreement with the Equation (3) and the set of parameters given in Table 8 for GPPS.

### 3.5. Prediction of Migration into Food and Food Simulants

Within this study, the diffusion coefficients were determined from permeation and migration experiments into the gas phase. Therefore, interaction of contact media like sorption and swelling of the polymer were minimized. Since swelling effects of the polymer matrix by sorption of the simulants is thereby negligible, the migration is independent of the simulant and the mass transfer is affected only by the diffusion of the migrant in the polymer. Interactions between contact media and polymers increase the migration which is an important parameter when using GPPS and HIPS in food packaging applications. The phenomena of sorption and swelling are well-known from migration kinetics into food simulants, e.g., the loss of functional barrier properties of PS functional barrier layers [32]. Strong interactions of PS with coconut oil, palm kernel oil, Miglyol, 10% ethanol, 50% ethanol and goat’s milk have been reported [15] in the case of polystyrene (in this publication, it is unclear if it is GPPS or HIPS), which indicate that these foods and simulants can swell the polymer matrix and increase the diffusion of potential migrants into food and simulants. A similar behavior is reported for acrylonitrile butadiene copolymer (ABS) [33]. Interaction between the polymer and the food simulants mainly influences the migration into food, which was also reported for PET [34]. However, real food does not swell the packaging polymer significantly, especially at low temperatures, otherwise packaging polymer might be unsuitable for this purpose. Swelling of food simulants compared to real food is shown for ABS in Lit. [33]. However, for convenience and as specified in regulatory guidelines, food compliance testing is typically performed with simulants like 50% ethanol or *iso*-octane. This leads to a strong over-estimation of the level of migration [33,34]. In addition, sorption and swelling had an influence also on the diffusion coefficients determined in PS. Genualdi et al. determined the diffusion coefficients of styrene and styrene oligomers into the food simulant 95% ethanol at 40 °C [35]. The diffusion coefficients are significantly higher as predicted from the results of this study, which is most probably due to the swelling effect of 95% ethanol as food simulant. The influence of swelling is also visible in the proposed prediction parameters, if migration data in swollen PS is taken into account [36]. In this recent compilation study, most of the diffusion coefficients were determined under swollen conditions, e.g., PS in neat solvents at high temperature in [37,38,39]. This leads to a strong over-estimation of the diffusion coefficients and therefore also to the derived modelling parameters. Realistic diffusion coefficients as derived from this study might lead to a more realistic prediction of the migration of polymer constituents into food.

## 4. Conclusions

Within this study, diffusion coefficients and activation energies of diffusion were determined experimentally from desorption kinetics into the gas phase. Activation energies of diffusion for a broad range of molecules below and above the glass transition temperature are not available in the scientific literature, which is the novelty of this study. From experimentally determined activation energies of diffusion, parameters for the prediction of the diffusion coefficients in GPPS and HIPS were established below and above the glass transition temperature of both polymers. These parameters *a* and *b* in Equation (3) were derived from the correlation between the pre-exponential factors D_0_ and the activation energies of diffusion E_A_. The parameters *c* and *d* were derived from the correlation between the activation energies of diffusion E_A_ and the molecular volume V of the investigated substances. From Equation (3) and the parameters *a* to *d* the diffusion coefficients D_P_ of other, non-tested organic substances can be predicted. In addition to the desorption kinetics, diffusion coefficients of the homologous row of 1-alcohols were determined by permeation tests through a thin GPPS film. The results of both methods, desorption and permeation kinetics, are in good agreement. Due to the fact that all diffusion coefficients were determined in gas phase kinetics, swelling effects and interactions between solvents and PS can be excluded. Therefore, the prediction parameters established within this study can be considered a pure diffusion coefficients. Most of the diffusion coefficients found in the scientific literature might include also swelling effects of the polymers with simulant media. Pure diffusion coefficients in polystyrene polymers might be useful for the understanding of the diffusion and migration processes. Migration modelling based on experimentally determined activation energies of diffusion, as established within this study, might therefore offer a basis for a more realistic estimation of the migration of polymer constituents into food.

## Figures and Tables

**Figure 1 polymers-13-01317-f001:**
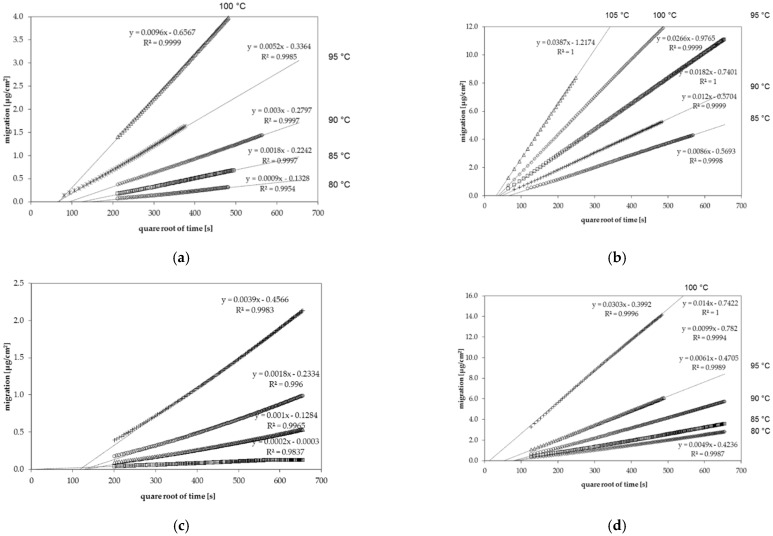
Gas phase migration kinetics of (**a**) *n*-octane from general purpose polystyrene (GPPS) at temperatures between 80 °C and 100 °C, (**b**) chlorobenzene from GPPS at temperatures between 85 °C and 105 °C, (**c**) *n*-dodecane from high impact polystyrene (HIPS) at temperatures between 80 °C and 100 °C and (**d**) toluene from HIPS at temperatures between 80 °C and 100 °C.

**Figure 2 polymers-13-01317-f002:**
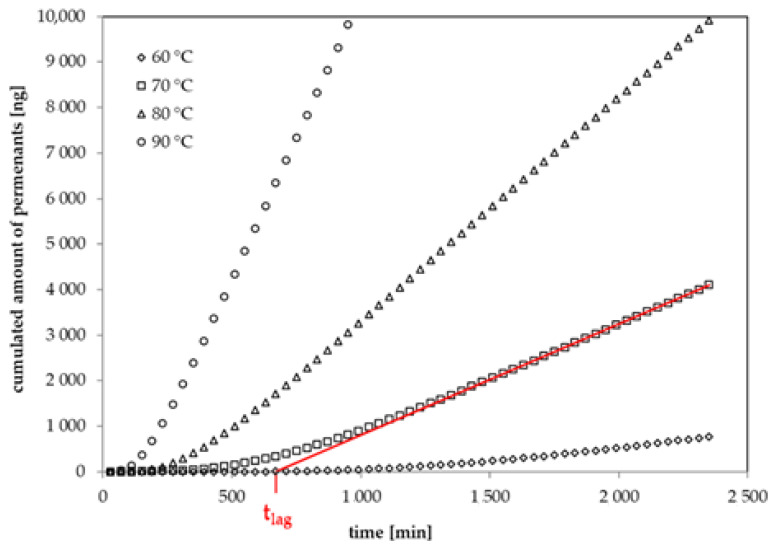
Experimental permeation curves for 1-butanol at 60 °C, 70 °C, 80 °C and 90 °C (lag time at 70 °C: intercept of the red line on the time-axis).

**Figure 3 polymers-13-01317-f003:**
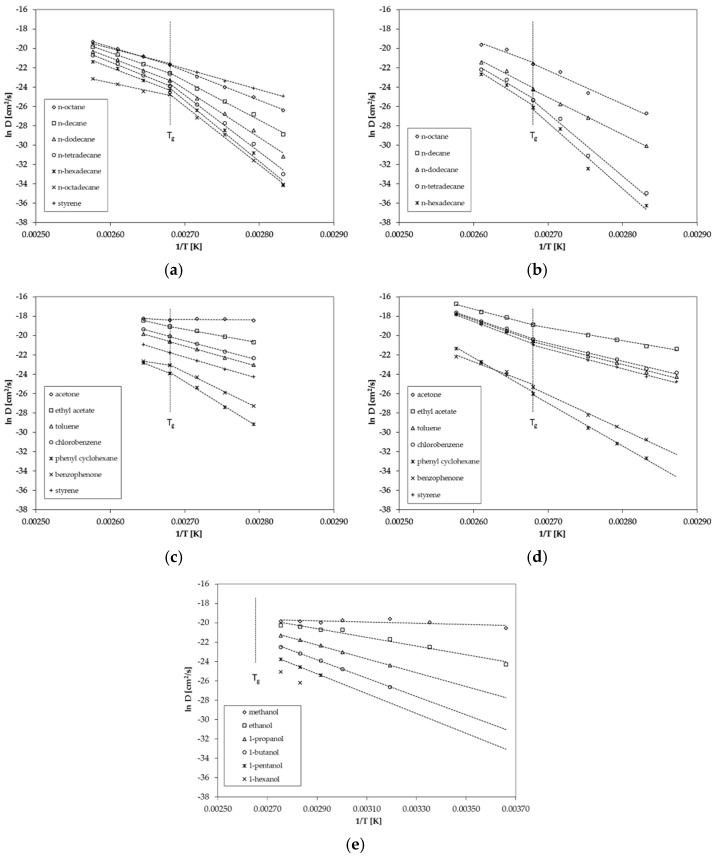
Correlation between ln D and reciprocal temperature (Arrhenius-plot) for (**a**) GPPS for *n*-alkanes, (**b**) HIPS for *n*-alkanes, (**c**) GPPS for non alkanes, (**d**) HIPS for non alkanes and (**e**) GPPS for 1-alcohols from the permeation tests.

**Figure 4 polymers-13-01317-f004:**
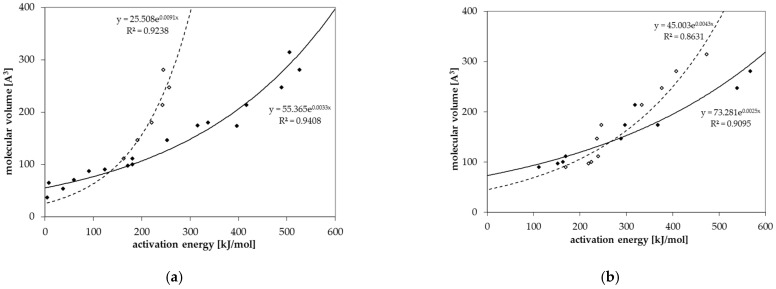
Correlation between the molecular volume V and the activation energy of diffusion E_A_ for (**a**) GPPS and (**b**) HIPS below T_g_ (solid dots) and above T_g_ (non-solid dots).

**Figure 5 polymers-13-01317-f005:**
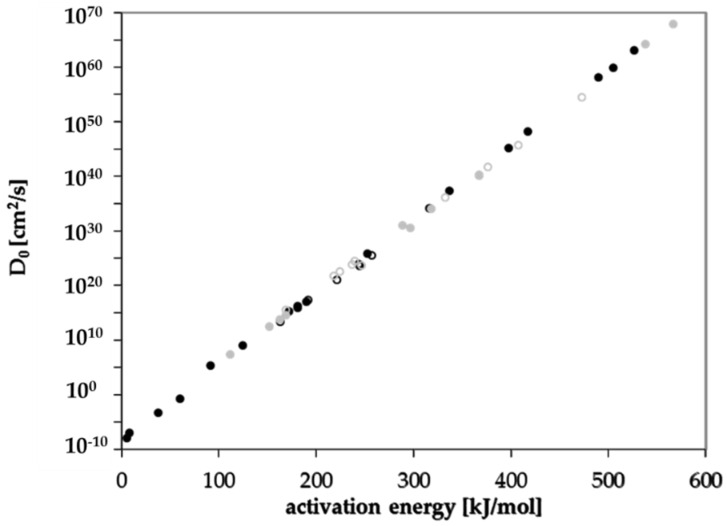
Correlation between the pre-exponential factor D_0_ and the activation energy of diffusion E_A_ below T_g_ (solid dots, solid line) and above T_g_ (non-solid dots, dashed line): black dots GPPS, grey dots HIPS.

**Table 1 polymers-13-01317-t001:** Experimentally determined concentrations of model compounds in the spiked polystyrene (PS) sheets.

Sheet	Substance	Spiked Concentration (mg/kg)	
		General Purpose Polystyrene (GPPS)	High Impact Polystyrene (HIPS)
sheet 1	*n*-Octane	422 ± 5	474 ± 37
	*n*-Decane	486 ± 4	354 ± 4
	*n*-Dodecane	518 ± 3	610 ± 8
	*n*-Tetradecane	531 ± 4	692 ± 15
	*n*-Hexadecane	538 ± 5	709 ± 19
	*n*-Octadecane	522 ± 5	674 ± 19
	Styrene ^1^	627 ± 22	354 ± 2
sheet 2	Acetone	180 ^2^	/
	Ethyl acetate	270 ^2^	500 ^2^
	Toluene	464 ± 3	763 ± 11
	Chlorobenzene	521 ± 4	821 ± 13
	Phenyl cyclohexane	627 ± 6	1229 ± 33
	Benzophenone	538 ± 7	1002 ± 27
	Styrene ^1^	595 ± 5	363 ± 6

^1^ residual monomer, not artificially added, ^2^ estimated from headspace gas chromatography.

**Table 2 polymers-13-01317-t002:** 1-Alcohols and their upstream concentrations used in permeation testing.

Temperature (°C)	Concentration (µg/L)					
	Methanol	Ethanol	1-Propanol	1-Butanol	1-Pentanol	1-Hexanol
0	9.16	12.8	9.29	7.49	5.65	3.76
25	27.5	28.3	27.9	22.5	17.0	11.3
40	54.9	76.7	55.8	45.0	33.9	45.0
60	201	281	204	165	124	82.8
70	330	460	335	270	203	135
80	549	767	558	450	339	226
90	916	1280	929	749	565	376

**Table 3 polymers-13-01317-t003:** Experimentally determined diffusion coefficients of *n*-alkanes and styrene in polystyrene GPPS and HIPS (sheet 1) from desorption kinetics.

Polymer	Temperature	Diffusion Coefficient (cm^2^/s)						
	(°C)	*n*-Octane	*n*-Decane	*n*-Dodecane	*n*-Tetradecane	*n*-Hexadecane	*n*-Octadecane	Styrene
GPPS	80	3.4 × 10^−12^	2.8 × 10^−13^	3.0 × 10^−14^	4.7 × 10^−15^	1.5 × 10^−15^	1.6 × 10^−15^	1.5 × 10^−11^
	85	1.4 × 10^−11^	2.2 × 10^−12^	4.3 × 10^−13^	1.1 × 10^−13^	4.1 × 10^−14^	1.9 × 10^−14^	3.3 × 10^−11^
	90	3.8 × 10^−11^	8.6 × 10^−12^	2.4 × 10^−12^	8.9 × 10^−13^	4.3 × 10^−13^	2.8 × 10^−13^	7.1 × 10^−11^
	95	1.1 × 10^−10^	3.3 × 10^−11^	1.2 × 10^−11^	6.3 × 10^−12^	3.4 × 10^−12^	1.6 × 10^−12^	1.8 × 10^−10^
	100	3.8 × 10^−10^	1.6 × 10^−10^	7.5 × 10^−11^	4.2 × 10^−11^	2.6 × 10^−11^	1.8 × 10^−11^	4.1 × 10^−10^
	105	8.6 × 10^−10^	4.0 × 10^−10^	2.1 × 10^−10^	1.3 × 10^−10^	7.6 × 10^−11^		9.7 × 10^−10^
	110	2.0 × 10^−9^	1.1 × 10^−9^	6.3 × 10^−10^	4.1 × 10^−10^	2.6 × 10^−10^		1.6 × 10^−9^
	115	4.0 × 10^−9^	2.4 × 10^−9^	1.5 × 10^−9^	9.9 × 10^−10^	5.2 × 10^−10^		3.3 × 10^−9^
HIPS	80	2.5 × 10^−12^	artefacts ^1^	8.6 × 10^−14^	6.7 × 10^−16^	1.8 × 10^−16^		
	90	2.0 × 10^−11^		1.6 × 10^−12^	3.1 × 10^−14^	8.1 × 10^−15^		
	95	1.8 × 10^−10^		6.5 × 10^−12^	1.4 × 10^−12^	5.0 × 10^−13^		
	100	4.0 × 10^−10^		3.0 × 10^−11^	9.7 × 10^−12^	4.6 × 10^−12^		
	105	1.9 × 10^−9^		2.0 × 10^−10^	8.2 × 10^−11^	4.6 × 10^−11^		
	110	2.9 × 10^−9^		4.9 × 10^−10^	2.3 × 10^−10^	1.4 × 10^−10^		

^1^ not determined due to analytical artefacts.

**Table 4 polymers-13-01317-t004:** Experimentally determined diffusion coefficients of *n*-alkanes and styrene in polystyrene GPPS and HIPS (sheet 2) from desorption kinetics.

Polymer	Temperature	Diffusion Coefficient (cm^2^/s)						
	(°C)	Acetone	Ethyl Acetate	Toluene	Chlorobenzene	Phenyl Cyclohexane	Benzophenone	Styrene
GPPS	85	9.8 × 10^−9^	1.1 × 10^−9^	1.0 × 10^−10^	2.0 × 10^−10^	2.1 × 10^−13^	1.4 × 10^−12^	2.9 × 10^−11^
	90	1.1 × 10^−8^	1.8 × 10^−9^	2.1 × 10^−10^	3.9 × 10^−10^	1.2 × 10^−12^	5.7 × 10^−12^	6.4 × 10^−11^
	95	1.2 × 10^−8^	3.3 × 10^−9^	4.9 × 10^−10^	8.8 × 10^−10^	9.5 × 10^−12^	2.8 × 10^−11^	1.5 × 10^−10^
	100	1.0 × 10^−8^	5.2 × 10^−9^	1.1 × 10^−9^	1.9 × 10^−9^	4.1 × 10^−11^	9.6 × 10^−11^	3.5 × 10^−10^
	105	1.2 × 10^−8^	9.8 × 10^−9^	2.4 × 10^−9^	4.0 × 10^−9^	1.2 × 10^−10^	1.5 × 10^−10^	8.3 × 10^−10^
HIPS	75	artefacts ^1^	5.1 × 10^−10^	2.9 × 10^−11^	4.5 × 10^−11^			1.8 × 10^−11^
	80		6.9 × 10^−10^	4.6 × 10^−11^	6.9 × 10^−11^	6.4 × 10^−15^	4.5 × 10^−14^	2.9 × 10^−11^
	85		1.3 × 10^−9^	1.2 × 10^−10^	1.7 × 10^−10^	4.0 × 10^−14^	1.7 × 10^−13^	7.9 × 10^−11^
	90		2.2 × 10^−9^	2.4 × 10^−10^	3.3 × 10^−10^	1.5 × 10^−13^	5.5 × 10^−13^	4.0 × 10^−11^
	100		6.3 × 10^−9^	1.1 × 10^−9^	1.4 × 10^−9^	5.1 × 10^−12^	1.0 × 10^−11^	8.2 × 10^−10^
	105		1.3 × 10^−8^	3.6 × 10^−9^	4.2 × 10^−9^	3.4 × 10^−11^	4.9 × 10^−11^	2.7 × 10^−9^
	110		2.3 × 10^−8^	7.8 × 10^−9^	8.7 × 10^−9^	1.4 × 10^−10^	1.1 × 10^−10^	6.2 × 10^−9^
	115		5.4 × 10^−8^	2.0 × 10^−8^	2.2 × 10^−8^	5.3 × 10^−10^	2.3 × 10^−10^	1.7 × 10^−8^

^1^ not determined due to analytical artefacts.

**Table 5 polymers-13-01317-t005:** Experimentally determined diffusion coefficients D_P_ in GPPS of 1-alcohols from permeation kinetics.

Temperature (°C)	Diffusion Coefficient (cm^2^/s)					
	Methanol	Ethanol	1-Propanol	1-Butanol	1-Pentanol	1-Hexanol
0	1.2 × 10^−9^	2.8 × 10^−11^				
25	2.2 × 10^−9^	1.7 × 10^−10^				
40	3.1 × 10^−9^	3.8 × 10^−10^	2.5 × 10^−11^	2.7 × 10^−12^		
60	2.7 × 10^−9^	1.0 × 10^−9^	1.0 × 10^−10^	1.7 × 10^−11^		
70	2.1 × 10^−9^	1.0 × 10^−9^	2.0 × 10^−10^	4.1 × 10^−11^	9.3 × 10^−12^	
80	2.4 × 10^−9^	1.4 × 10^−9^	3.5 × 10^−10^	8.5 × 10^−11^	2.1 × 10^−11^	4.2 × 10^−12^
90	2.4 × 10^−9^	1.6 × 10^−9^	5.6 × 10^−10^	1.7 × 10^−10^	4.8 × 10^−11^	1.3 × 10^−11^

**Table 6 polymers-13-01317-t006:** Activation energies of diffusion E_A_ and pre-exponential factors D_0_ for GPPS.

Substance	Molecular Volume	Molecular Weight	Temperature Range	Activation Energy	Pre-Exponential Factor D_0_	Method
	(Å^3^)	(g/mol)	(°C)	(kJ/mol)	(cm^2^/s)	
Methanol	37.2	32.0	0–90	4.9	1.4 × 10^−8^	Permeation, below T_g_
Ethanol	54.0	46.1	0–90	37.3	5.1 × 10^−4^	Permeation, below T_g_
Acetone	64.7	58.1	85–105	7.6	1.3 × 10^−7^	Desorption
1-Propanol	70.8	60.1	40–90	59.6	2.2 × 10^−1^	Permeation, below T_g_
1-Butanol	87.6	74.1	40–90	91.1	2.6 × 10^5^	Permeation, below T_g_
Ethyl acetate	90.6	88.1	85–105	123.9	1.2 × 10^9^	Desorption
Chlorobenzene	97.6	112.6	85–105	171.3	1.8 × 10^15^	Desorption
Toluene	100.6	92.1	85–105	180.6	2.1 × 10^16^	Desorption
Styrene	111.8	104.2	80–100	180.9	8.2 × 10^15^	Desorption, below T_g_
Styrene	111.8	104.2	100–115	162.6	2.5 × 10^13^	Desorption, above T_g_
Styrene	111.8	104.2	85–105	189.8	1.3 × 10^17^	Desorption
*n*-Octane	146.6	114.2	80–100	252.1	7.0 × 10^25^	Desorption, below T_g_
*n*-Octane	146.6	114.2	100–115	191.3	2.3 × 10^17^	Desorption, above T_g_
Phenyl cyclohexane	174.0	160.3	85–105	396.8	1.6 × 10^45^	Desorption
Benzophenone	174.4	182.2	85–105	315.5	1.5 × 10^34^	Desorption
*n*-Decane	180.2	142.3	80–100	336.7	2.2 × 10^37^	Desorption, below T_g_
*n*-Decane	180.2	142.3	100–115	220.6	1.2 × 10^21^	Desorption, above T_g_
*n*-Dodecane	213.8	170.3	80–100	416.4	1.6 × 10^48^	Desorption, below T_g_
*n*-Dodecane	213.8	170.3	100–115	243.1	8.1 × 10^23^	Desorption, above T_g_
*n*-Tetradecane	247.3	198.4	80–100	489.1	1.5 × 10^58^	Desorption, below T_g_
*n*-Tetradecane	247.3	198.4	100–115	256.4	3.4 × 10^25^	Desorption, above T_g_
*n*-Hexadecane	281.0	226.5	80–100	525.8	1.4 × 10^63^	Desorption, below T_g_
*n*-Hexadecane	281.0	226.5	100–115	244.3	4.4 × 10^23^	Desorption, above T_g_
*n*-Octadecane	314.6	254.5	80–100	504.9	8.5 × 10^59^	Desorption, below T_g_

**Table 7 polymers-13-01317-t007:** Activation energies of diffusion E_A_ and pre-exponential factors D_0_ for HIPS.

Substance	Molecular Volume	Molecular Weight	Temperature Range	Activation Energy	Pre-Exponential Factor D_0_	Method
	(Å^3^)	(g/mol)	(°C)	(kJ/mol)	(cm^2^/s)	
Ethyl acetate	90.6	88.1	75–100	111.4	2.4 × 10^7^	Desorption, below T_g_
Ethyl acetate	90.6	88.1	100–115	169.3	3.1 × 10^15^	Desorption, above T_g_
Chlorobenzene	97.6	112.6	75–100	152.2	2.6 × 10^12^	Desorption, below T_g_
Chlorobenzene	97.6	112.6	100–115	218.1	5.1 × 10^21^	Desorption, above T_g_
Toluene	100.6	92.1	75–100	162.8	6.7 × 10^13^	Desorption, below T_g_
Toluene	100.6	92.1	100–115	224.4	3.2 × 10^22^	Desorption, above T_g_
Styrene	111.8	104.2	75–100	169.2	3.7 × 10^14^	Desorption, below T_g_
Styrene	111.8	104.2	100–115	239.3	6.9 × 10^23^	Desorption, above T_g_
*n*-Octane	146.6	114.2	80–100	288.5	1.0 × 10^31^	Desorption, below T_g_
*n*-Octane	146.6	114.2	100–110	236.8	6.7 × 10^23^	Desorption, above T_g_
Phenyl cyclohexane	174.0	160.3	80–100	367.4	1.2 × 10^40^	Desorption, below T_g_
Phenyl cyclohexane	174.0	160.3	100–115	367.7	1.8 × 10^40^	Desorption, above T_g_
Benzophenone	174.4	182.2	80–100	297.0	3.4 × 10^30^	Desorption, below T_g_
Benzophenone	174.4	182.2	100–115	246.3	3.9 × 10^23^	Desorption, above T_g_
*n*-Dodecane	213.8	170.3	80–100	318.7	1.4 × 10^34^	Desorption, below T_g_
*n*-Dodecane	213.8	170.3	100–115	233.0	1.4 × 10^36^	Desorption, above T_g_
*n*-Tetradecane	247.3	198.4	80–100	538.4	2.1 × 10^64^	Desorption, below T_g_
*n*-Tetradecane	247.3	198.4	100–110	376.1	5.1 × 10^41^	Desorption, above T_g_
*n*-Hexadecane	281.0	226.5	80–100	567.2	9.3 × 10^67^	Desorption, below T_g_
*n*-Hexadecane	281.0	226.5	100–110	407.6	6.3 × 10^45^	Desorption, above T_g_
*n*-Octadecane	314.6	254.5	100–110	472.8	3.7 × 10^54^	Desorption, above T_g_

**Table 8 polymers-13-01317-t008:** Parameters for the prediction of diffusion coefficients according to Equation (3) for GPPS and HIPS below and above Tg.

Parameter	GPPS		HIPS	
	below T_g_	above T_g_	below T_g_	above T_g_
a (1/K)	2.59 × 10^−3^	2.44 × 10^−3^	2.55 × 10^−3^	2.46 × 10^−3^
b (cm^2^/s)	7.38 × 10^−9^	6.46 × 10^−8^	9.21 × 10^−9^	2.07 × 10^−7^
c (Å^3^)	55.71	25.51	73.28	45.00
d (1/K)	2.73 × 10^−5^	7.55 × 10^−5^	2.04 × 10^−5^	3.57 × 10^−5^

## Data Availability

The data presented in this study are available on request from the corresponding author.

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
