# Peer review of "Diffusion Coefficients and Activation Energies of Diffusion of Organic Molecules in Polystyrene below and above Glass Transition Temperature"

_polymers, 2021, doi:10.3390/polym13081317_

Round 1

Reviewer 1 Report

This work reports the diffusion coefficients of organic molecules in GPPS and HIPS over a broad temperature range below and above the glass transition temperature. Desorption kinetics into the gas phase with spiked GPPS and HIPS sheets and permeation kinetics through a thin GPPS film were studied and then diffusion coefficients were determined. The activation energies of diffusion EA and pre-exponential factor D0 were obtained from the temperature dependency of the diffusion coefficients. A prediction equation for diffusion coefficients was established. This work is of great significance to the prediction of barrier properties of PS food packaging materials. The manuscript is well-presented and the data is discussed appropriately. I recommend this manuscript to be accepted. Some comments are as follows.

All tested compounds follow the correlation between the EA and V nearly independent from chemical nature, functional groups or polarity of the molecules. This is the case for the PS matrix, which is a non-polar matrix. What is the difference if the polymer is a polar matrix. In that case, except the penetrant volume, some other factors such as the interaction force between the penetrant and the polymer matrix,  may also affect the EA.  

Author Response

Reviewer 1

This work reports the diffusion coefficients of organic molecules in GPPS and HIPS over a broad temperature range below and above the glass transition temperature. Desorption kinetics into the gas phase with spiked GPPS and HIPS sheets and permeation kinetics through a thin GPPS film were studied and then diffusion coefficients were determined. The activation energies of diffusion EA and pre-exponential factor D0 were obtained from the temperature dependency of the diffusion coefficients. A prediction equation for diffusion coefficients was established. This work is of great significance to the prediction of barrier properties of PS food packaging materials. The manuscript is well-presented and the data is discussed appropriately. I recommend this manuscript to be accepted. Some comments are as follows.

All tested compounds follow the correlation between the EA and V nearly independent from chemical nature, functional groups or polarity of the molecules. This is the case for the PS matrix, which is a non-polar matrix. What is the difference if the polymer is a polar matrix. In that case, except the penetrant volume, some other factors such as the interaction force between the penetrant and the polymer matrix,  may also affect the EA. 

Answer FW: Thanks for this comment: The diffusion coefficients and therefore also the activation energy is only (or mainly) influenced by the molecular size. This was found also for polar matrices for the same set of substances e.g. PET, which is a polar polymer (Journal of Applied Polymer Science, 2013, 129(4), 1845-1851). Polarity and functional groups play a major role in the partitioning coefficient (Functional barrier performance of a polyamide 6 membrane towards n-alkanes and 1-alcohols, Packaging Technology and Science, 2016, 29(6), 277-287). I introduced a short discussion on this topic into the manuscript.

Reviewer 2 Report

Dear Author,

This an interesting paper, including accurate and systematic work with valuable information for those in the field of food packgainf area. Some comments and suggestions are enclosed for improving paper quality before been considered for publication.

Abstract.

Please remove the words, background, methods, and results from it.

As a general consideration in the whole manuscript, recent publications must be included. There are few ones from the last 5 years.

Methods section.

It is important to add all chemical suppliers and PS main chemical properties.

Results section

Is there any picture and physicochemical characterization of the sheets used that can be added to the manuscript?

Why n-octane and styrene were not reported in Table 3, and acetone in Table 4? Why always you find issues with the same polymer? Would it depend on the physicochemical properties of the polymer sheets used?

Conclusions

I suggest providing a shorter conclusion, some parts of it can be included in the previous section. 

References

Suggested included updated references.

Author Response

Reviewer 2

This an interesting paper, including accurate and systematic work with valuable information for those in the field of food packgainf area. Some comments and suggestions are enclosed for improving paper quality before been considered for publication.

Abstract.

Please remove the words, background, methods, and results from it.

Answer FW: Thanks for your comments. I removed it from the abstract.

As a general consideration in the whole manuscript, recent publications must be included. There are few ones from the last 5 years.

Answer FW: I made a literature search and added five paper. I focussed on Papers, which mentioned diffusion coefficients. There are more paper dealing with the diffusion/migration of styrene only. I excluded these paper, because this is not the focus of the manuscript. The focus of this paper are on activation energies of diffusion of a broad range of molecules.

Methods section.

It is important to add all chemical suppliers and PS main chemical properties.

Answer FW: Due to the fact, that lots of different chemicals have been used in the study, I decided not to include the chemical suppliers of the chemicals and standard substances. In some cases, the chemicals for the spiking experiments were from different suppliers as for the analytical standards. This would be confusing for the reader. Hope this is acceptable for the Journal.

Results section

Is there any picture and physicochemical characterization of the sheets used that can be added to the manuscript?

Answer FW: The sheets were only characterized by extraction of the material in order to determine the concentrations of the spiked molecules. During these tests, also the polystyrene oligomers were determined. The concentrations of the oligomers are similar in the spiked sheets, in the non-spiked references sheets as well as in the pellets.

Why n-octane and styrene were not reported in Table 3, and acetone in Table 4? Why always you find issues with the same polymer? Would it depend on the physicochemical properties of the polymer sheets used?

Answer FW: Both substances cannot be determined properly due to analytical artefacts. The sheets were spiked with very low molecular weight substances like acetone as well as high molecular weight substances up to tetracosane. So we have to apply a GC method which is able the analyse the complete molecular weight range.  

Conclusions

I suggest providing a shorter conclusion, some parts of it can be included in the previous section.

Answer FW: I added a new chapter and I shorten the conclusions.

References

Suggested included updated references.

Answer FW: Five new references are included. Se above.

Reviewer 3 Report

Frank Welle reported the manuscript entitled ‘‘Diffusion Coefficients and Activation Energies of Diffusion of Organic Molecules in Polystyrene below and above Glass Transition Temperature’’. This manuscript needs some minor revision before publication. 

  1. Abstract should be more quantitative.
  2. The introduction section should be a comparative study with polystyrene-based (recently) articles. Please mention the novelty of this work.
  3. Please add a new paragraph regarding statistical analysis and clearly explain the number of repeats, replications, level of significance, software to analyze the data.
  4. Please improve the quality of figure 1.
  5. If possible please compare activation energy calculated by an Arrhenius approach with the other approach.

Author Response

Reviewer 3

Frank Welle reported the manuscript entitled ‘‘Diffusion Coefficients and Activation Energies of Diffusion of Organic Molecules in Polystyrene below and above Glass Transition Temperature’’. This manuscript needs some minor revision before publication.

    Abstract should be more quantitative.

Answer FW: Thanks for your comments. The Journals template allows only 200 words in the abstract. So I had to focus on what has been done in the study, not on the results.

    The introduction section should be a comparative study with polystyrene-based (recently) articles. Please mention the novelty of this work.

Answer FW: I made a literature search and added five paper. I focussed on Papers, which mentioned diffusion coefficients. There are more paper dealing with the diffusion/migration of styrene only. I excluded these paper, because this is not the focus of the manuscript.

Answer FW: The novelty of the work was mentioned in the conclusions section.

    Please add a new paragraph regarding statistical analysis and clearly explain the number of repeats, replications, level of significance, software to analyze the data.

Answer FW: We analysed the kinetics only ones per sheet and temperature. I mentioned this in the experimental section. So, standard deviations cannot be given. However, from the comparison with different temperatures, it can be concluded, that the tests are reproducible. This however, results in the activation energies of diffusion, which are discussed in detail in the manuscript. The kinetics were evaluated in excel. No special software was used.  

    Please improve the quality of figure 1.

Answer FW: I think, this was only a wrong page break. Figure 1 has been improved.

    If possible please compare activation energy calculated by an Arrhenius approach with the other approach.

Answer FW: In the food packaging area, the diffusion models are using only the Arrhenius approach, because it is mentioned in EU Regulation 10/2011. I therefore made no changes in the manuscript regarding this recommendation.

Round 2

Reviewer 2 Report

No further comments on this paper. I suggest accepting in present form.